# Pyrolytic Graphite for an In-Plane Force Study of Diamagnetic Levitation: A Potential Microdetector of Cracks in Magnetic Material

**DOI:** 10.3390/mi14061242

**Published:** 2023-06-13

**Authors:** Runze Liu, Wenjiang Yang, Hongjun Xiang, Peng Zhao, Fuwen Deng, Juzhuang Yan

**Affiliations:** School of Astronautics, Beihang University, Beijing 100191, China; liurunze2000@buaa.edu.cn (R.L.); hjxiang@263.net (H.X.); zy2015329@buaa.edu.cn (P.Z.); sy2115305@buaa.edu.cn (F.D.); ju1328874221@163.com (J.Y.)

**Keywords:** pyrolytic graphite, diamagnetic levitation, cracks detector, in-plane motion, fixed-axis rotation

## Abstract

The diamagnetic levitation technique can be applied in non-destructive testing for identifying cracks and defects in magnetic materials. Pyrolytic graphite is a material that can be leveraged in micromachines due to its no-power diamagnetic levitation on a permanent magnet (PM) array. However, the damping force applied to pyrolytic graphite prevents it from maintaining continuous motion along the PM array. This study investigated the diamagnetic levitation process of pyrolytic graphite on a permanent magnet array from various aspects and drew several important conclusions. Firstly, the intersection points on the permanent magnet array had the lowest potential energy and validated the stable levitation of pyrolytic graphite on these points. Secondly, the force exerted on the pyrolytic graphite during in-plane motion was at the micronewton level. The magnitude of the in-plane force and the stable time of the pyrolytic graphite were related to the size ratio between it and the PM. During the fixed-axis rotation process, the friction coefficient and friction force decreased as the rotational speed decreased. Smaller-sized pyrolytic graphite can be used for magnetic detection, precise positioning and other microdevices. The diamagnetic levitation of pyrolytic graphite can also be used for detecting cracks and defects in magnetic materials. We hope this technique will be used in crack detection, magnetic detection and other micromachines.

## 1. Introduction

Pyrolytic graphite is a unique material that exhibits strong diamagnetic properties, allowing it to stably levitate on a permanent magnet array without an external energy input. The advantage of diamagnetic levitation over other methods using magnetic forces is that diamagnetic objects can be stably levitated without active control at room temperature [1]. Thus, pyrolytic graphite can be used for magnetic field detection, precise positioning, micromachinery [2,3], nanorobots [4] and susceptible sensors [5] due to its advantages of passive and non-contact diamagnetic levitation. Cheng [6] experimented on an energy harvester based on a diamagnetic levitation structure. Ding [7] used pyrolytic graphite to fabricate rotors, suggesting potential applications of this mechanism in suspension bearings and microactuators. Masayuki [8] used graphite, Nd-Fe-B permanent magnets and light to create an optical motion control system. Zhang [9] studied a gas flowmeter using a diamagnetic levitation rotor made of pyrolytic graphite. Ali [10] used pyrolytic graphite to design three control models that took the hydrodynamic effects exerted on the microrobot into account. Hüseyin [11] proposed a novel microrobotic manipulation technique with high precision (nanoscale) positioning capability, which was suitable for movement in an environment featuring the diamagnetic levitation of liquid.

In this study, we aimed to apply the pyrolytic graphite levitation technique as a non-destructive test for identifying cracks and defects in magnetic materials, as shown in Figure 1. A stable magnetic field used for pyrolytic graphite was generated by a PM array, which was constructed using multiple small magnets arranged in intersecting points and lines. The occurrence of surface or internal damage or cracks in permanent magnets alters their magnetic field. The pyrolytic graphite generates an eddy current damping effect as it approaches the crack due to a sudden alteration in the magnetic field, altering its in-state of planar motion, thereby facilitating precise identification of the crack’s location. This method is valuable, as it enables non-contact testing that minimizes the risk of damaging the tested materials, improves the test’s efficiency and reduces the overall costs.

Despite the huge potential offered by pyrolytic graphite magnetic levitation technology, its practical applications are restricted by the phenomenon of in-plane forces acting on the in-plane motion. Previous studies have often regarded pyrolytic graphite as the substrate for magnetic levitation and permanent magnets as the carrier for in-plane motion. Instead, in our research, pyrolytic graphite was used as the carrier of in-plane motion. This study aimed to investigate the characteristics of the in-plane motion of the pyrolytic graphite plate, thereby determining the distribution of the magnetic potential energy generated and analyzing the position of levitation, the in-plane forces and the damping ratio of pyrolytic graphite during in-plane motion, as well as the level of frictional force and the friction coefficient during its rotational motion around a fixed axis.

## 2. Material and Methods

Earnshaw’s theorem states that charged objects cannot be maintained in a stable stationary equilibrium solely by the electrostatic interaction of the charges [12]. The diamagnetism of pyrolytic graphite is caused by Landau diamagnetism, which is closely related to the unique Dirac cone band structure and the Fermi level of graphite [13]. When an external magnetic field is applied, low-energy electrons will migrate from the valence band to the conduction band. The mobility of electrons with a high magnetic flux will generate a robust equivalent current and then create a reverse-induced magnetic field to resist changes in the external magnetic field. Since the direction of the equivalent magnetic moment is different from the outside, pyrolytic graphite exhibits diamagnetism [14]. 

The magnetic force *F* is [15]:(1)$$F=\int_{V}^{}fdV$$

The force density *f* can be expressed as:(2)$$f=M\cdot \nabla \left(B\right)$$

For the diamagnetic materials
(3)$$H=\frac{B}{\mu_{0}}-M$$
(4)M=χH
(5)$$M=\frac{\chi B}{\mu_{0}(1+\chi )}\approx \frac{1}{\mu_{0}}\chi {\cdot B}_{}^{}$$
where *H* is the magnetic field strength, *B* is the magnetic flux density, *M* is the magnetization, *χ* is the magnetic susceptibility and *μ*_0_ is the vacuum permeability (*μ*_0_ = 4π × 10^−7^).

The final expression for the force density *f* is
(6)$$f=\frac{1}{{2\mu }_{0}}\chi \cdot \nabla \left(B_{}^{2}\right)=\frac{1}{2\mu_{0}}\nabla \left(\chi_{x}^{2}B_{x}^{2}+\chi_{y}^{2}B_{y}^{2}+\chi_{z}^{2}B_{z}^{2}\right)$$
where *χ_x_* is the magnetic susceptibility along the X-direction, *χ_y_* is the magnetic susceptibility along the Y-direction and *χ_z_* is the magnetic susceptibility along the Z-direction.

Simon and Geim have determined the value of *χ* for PG, which is given by [16]:(7)$$\chi =-\left[\begin{array}{ccc} 85 & 0 & 0\\ 0 & 85 & 0\\ 0 & 0 & 450 \end{array}\right]\times 10^{-6}$$

The magnetic force of the entire volume can be expressed as:(8)$$F=∰\frac{1}{{2\mu }_{0}}\chi \cdot \nabla \left(B_{}^{2}\right)dv$$

The force along the y-direction of the entire volume is represented as [4]
(9)$$F_{y}=\frac{1}{{2\mu }_{0}}∰\left(\chi_{x}^{}\frac{{\partial B}_{x}^{2}}{\partial y}+\chi_{y}^{}\frac{{\partial B}_{y}^{2}}{\partial y}+\chi_{z}^{}\frac{{\partial B}_{z}^{2}}{\partial y}\right)dV$$

## 3. Characteristics of Location

To begin with, we evaluated the distribution of the magnetic flux density in *X*, *Y* and *Z* directions above the PM array, as shown in Figure 2, where the total magnetic flux density is $B=\sqrt{B_{x}^{2}+B_{y}^{2}+B_{z}^{2}}$. The magnetic flux density is mainly distributed at the intersecting line of the PM array. The further away it is from the intersecting line of the PM array, the lower the magnetic flux density.

Then we investigated the potential energy distribution above the PM array to determine the position of stable levitation of pyrolytic graphite. The total potential energy above the PM array can be calculated using the following equation [17]:(10)$$U=-\int_{V}^{}M\cdot BdV$$

When the pyrolytic graphite stably levitates above the PM array, it should be located at the lowest point of potential energy in this region [18]. If the position of pyrolytic graphite cannot reach the lowest point of local potential energy, it cannot levitate stably, resulting in translation, vibration, rotation and other motions. The diamagnetic potential of pyrolytic graphite can be expressed as follows [17]
(11)$$U=-\frac{VB_{}^{2}\chi_{z}}{2\mu_{0}}$$
where *V* is the volume of graphite. 

The distribution of the potential energy above the PM array can be obtained as depicted in Figure 3. It is evident that the potential energy reached its maximum at the intersecting line of the PM arrays and its minimum at the intersecting point. This observation suggested that the centroid of pyrolytic graphite will be levitated in a stable manner above the intersection point of the PM arrays.

## 4. Characteristics of In-Plane Motion

The experimental system for in-plane motion is shown in Figure 4. Initially, a gentle air flow was used to drive the pyrolytic graphite. Simultaneously, a camera above the PM array transmitted real-time video of the pyrolytic graphite’s movements to a PC. Subsequently, the PC was utilized to identify the contour (blue rectangle in Figure 4) and centroid (red dot in Figure 4) of the pyrolytic graphite from the captured video footage. The coordinates of the displacement of the centroid in each frame can be determined, which can then be used to calculate the centroid’s velocity, acceleration and other parameters associated with its in-plane motion using Formulas (12) and (13)
(12)$$V(k)=\frac{\sqrt{{\left[Cent\left(k,1\right)-Cent\left(k-1,1\right)\right]}^{2}+{\left[Cent\left(k,2\right)-Cent\left(k-1,2\right)\right]}^{2}}}{∆t}\times \frac{D}{DIAM}$$
(13)$$a(k)=\frac{V\left(k\right)-V(k-1)}{∆t}$$
where *V (k)* is the velocity of centroid in frame *K*; *Cent* (*k*, 1) and *Cent* (*k −* 1, 1) are the X coordinates corresponding to the centroid of pyrolytic graphite in frames *K* and *K* − 1, respectively; *Cent* (*k*, 2) and *Cent* (*k* − 1, 2) are the Y coordinates corresponding to the centroid of pyrolytic graphite in frames *K* and *K* − 1, respectively; ∆*t* is the time interval; *D* is the actual side length of the pyrolytic graphite; *DIAM* is the pixel length corresponding to the calibration line and *a (k)* is the acceleration of the pyrolytic graphite’s motion in frame *K*. These motion parameters can efficiently calculate the in-plane force acting on the pyrolytic graphite.

The positional characteristics of pyrolytic graphite during in-plane oscillation were also examined. In this experiment, we used pyrolytic graphite with dimensions of 5 × 5 × 0.5 mm^3^ (PG5). Firstly, a baseline was drawn on the PM array, and its two endpoints were labeled as A and B. To eliminate the impact of the video frame rate on the parameters of PG5, we released the PG5 at the same location and captured the motion parameters of its centroid using two different video frame rates (30 fps and 60 fps). Due to the in-plane force, the oscillation displacement of the pyrolytic graphite gradually decreased and ultimately stabilized at a specific point. The displacement between the centroid of the pyrolytic graphite and Point A in the X-direction was determined, as depicted in Figure 5. The analysis revealed that the centroid of the pyrolytic graphite will levitate at the intersection point of the PM array.

## 5. Damping Characteristics

### 5.1. In-Plane Motion

In this experiment, PG10 (10 × 10 × 0.5 mm^3^) and PG20 (20 × 20 × 0.5 mm^3^) were used to investigate the in-plane force. The outcomes shown in Figure 6 and Figure 7 indicated that PG10 demonstrated a greater acceleration and stabilization rate compared with PG20. The in-plane force *F* can be calculated by
(14)F=ρmnh·a
where *ρ* = 2.2 g/cm^3^ is the density of the PG; *m* and *n* are the length and width of the PG, respectively; *h* is the thickness of the PG and *a* is the acceleration of the PG. The in-plane force on PG10 and PG20 was about 61.4 μN and 28.2 μN, respectively. This means that the in-plane force was at the micronewton level. Interestingly, PG10 reached stability faster and received a greater in-plane force than PG20.The smaller ratio of the size of GP10 to the permanent magnet compared with GP20 led to more uniform in-plane forces being experienced by PG20. Accordingly, we propose that the duration and magnitude of the in-plane force stability of pyrolytic graphite are dependent on the size ratio of the PG to the PM array. Specifically, larger pyrolytic graphite experiences diminished in-plane forces and can achieve extended periods of stable levitation, ultimately resulting in smoother and stabler motion. As the size of the pyrolytic graphite increases, the in-plane force is distributed over a larger area, leading to reduced forces per unit of area and greater stability. 

We also examined the damping ratio of pyrolytic graphite during in-plane oscillation. The in-plane force caused a gradual reduction in the oscillation displacement of pyrolytic graphite. PG5 and PG10 were used to investigate this phenomenon. The centroid displacement curves in the X-direction were obtained by releasing the pyrolytic graphite at the same point. Figure 8 displays the displacement curves in the X-direction for PG5.

The damping ratio *ξ* is calculated by [19]:(15)$$\xi =\frac{ln\frac{X\left(A\right)}{X\left(B\right)}}{2\pi \left(B-A\right)\sqrt{1+{\left[\frac{1}{2\pi \left(B-A\right)}ln\frac{X\left(A\right)}{X\left(B\right)}\right]}^{2}}}$$

The values *X*(*A*) and *X*(*B*) from the centroid’s displacement curve represent the peak and trough values, respectively, with A and B corresponding to the times of motion. The damping ratios for PG5 and PG10 were then calculated and are presented in Table 1. The damping ratio during in-plane oscillation was found to be between 0.1 and 0.2. The damping ratio of PG5 was lower than that of PG10, indicating an increase in the damping ratio with an increase in the size of the PG. We believe that the increase in the damping ratio with a variation in the size may be related to the increase in the surface area and mass, leading to more significant dissipation of energy. Furthermore, smaller pyrolytic graphite has a lower moment of inertia and can move more quickly. This suggests that smaller pyrolytic graphite would be suitable for the precise positioning of micromachines.

### 5.2. Fixed-Axis Rotation

During the process of magnetic levitation, pyrolytic graphite undergoes not only in-plane translational motion, but also in-plane fixed-axis rotational motion. When pyrolytic graphite undergoes fixed-axis rotation, the rotational friction force hinders its motion. Therefore, this part investigated the force and coefficient of friction ofPG10 during fixed-axis rotation using a polarity-crossing annular PM array and a white sticker to cover half of the PG10’s surface. The experimental setup for fixed-axis rotation is shown in Figure 9, where the rotational speed and acceleration were obtained by scale conversion, as represented in Figure 10 and Figure 11. To remove noise, it was necessary to filter the data of the fixed-axis rotational speed of graphite. The de-noised rotational speed is depicted by the red line in Figure 10, while the blue line represents the original dataset.

As shown in Figure 10, the stable rotation stage occurred for 0–15 s. Over time, the rotational speed gradually decreased, and the velocity and acceleration curves exhibited a relatively smooth trend. The stage of oscillating rotation occurred at approximately 15–30 s, when pyrolytic graphite exhibited an unstable rotation pattern characterized by oscillations rather than a stable cycle. After 30 s, the motion of pyrolytic graphite tended to stabilize, and PG10 eventually achieved stable levitation on the circular PM array.

The force and coefficient of friction are calculated by
(16)$$F_{\mu }=\mu m_{10}g=m_{10}a_{\mu }$$
where *F_μ_* is the friction of PG, *μ* is the coefficient of friction, *m*_10_ is the mass of PG10, *g* is the acceleration of gravity (*g* = 9.8 m/s^2^) and *a_μ_* is the acceleration of PG10. 

The calculated results indicated that the magnitude of the coefficient of friction was 10^−3^. When PG10 was in a stable rotation stage, the maximum frictional force was approximately 0.8 μN. It can be seen from Figure 11 that as the rotational speed decreased, the frictional force gradually decreased. We believe that the reason for this phenomenon may be related to air resistance because pyrolytic graphite with a higher rotational speed of means that air resistance consumes more power in a unit of time, which accelerates the rate of the reduction in speed. In addition, as previously mentioned, magnetic levitation technology has the advantage of being non-contact. In this experiment, only a momentary airflow was applied to the pyrolytic graphite. If a long-term and stable airflow can be obtained, pyrolytic graphite could be expected to be used as a gas bearing or a gas flow meter.

## 6. Results and Discussion

This study investigated the diamagnetic levitation process of pyrolytic graphite on a permanent magnet array from various aspects and drew several important conclusions. First, we found that the intersection points on the permanent magnet array had the lowest potential energy and validated the stable levitation of pyrolytic graphite at this point. Second, we discovered that the force exerted on the pyrolytic graphite during in-plane motion was at the micronewton level, and the damping ratio during oscillation was between 0.1 and 0.2. We believe that the magnitude of the in-plane force and the stable time of the pyrolytic graphite are related to the size ratio between it and the PM. With an increase in the size of the pyrolytic graphite, the force acting on it will be distributed over a larger area, reducing the force on each individual point and promoting the stability of its motion. Lastly, during the process of fixed-axis rotation, the coefficient of friction and friction force decreased as the rotational speed decreased. 

These results demonstrated that these forces impede pyrolytic graphite’s in-plane motion. There are two primary sources of in-plane force: air resistance and eddy current damping. During the in-plane movement of pyrolytic graphite, air generates a frictional force that opposes the motion. Chen [20] modeled eddy current damping and determined that it dominates dissipation in mm-sized pyrolytic graphite. On the basis of previous research and these findings, we posit that the in-plane force is the macroscopic manifestation of the combined effects of air resistance and eddy current damping.

We expect to improve the stability of magnetic levitation of pyrolytic graphite, prolong the in-plane motion time and ultimately enable the use of pyrolytic graphite for magnetic detection, precise positioning and other micromachines. For example, small-sized pyrolytic graphite can be used for fast and accurate positioning. In addition, the in-plane motion characteristics of pyrolytic graphite as a magnetic levitation material can also be applied in the field of magnetic sensors. For instance, this material can be used in high-precision magnetic sensors to achieve higher measurement accuracy and a wider measurement range. Table 2 presents the potential applications of pyrolytic graphite with different size specifications using diamagnetic levitation technology. In summary, combining the characteristics of pyrolytic graphite as a magnetic levitation material with magnetic detection technology could have various applications, providing strong support for future scientific research and industrial applications. Further research is expected to be conducted to fully exploit the potential applications of pyrolytic graphite in the field of micromachines.

## Figures and Tables

**Figure 1 micromachines-14-01242-f001:**
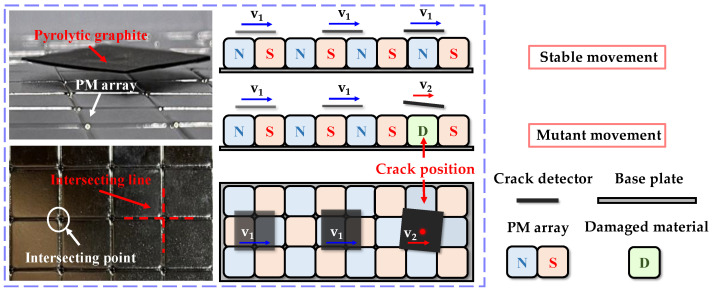
Pyrolytic graphite levitation technique for non-destructive testing of magnetic materials.

**Figure 2 micromachines-14-01242-f002:**
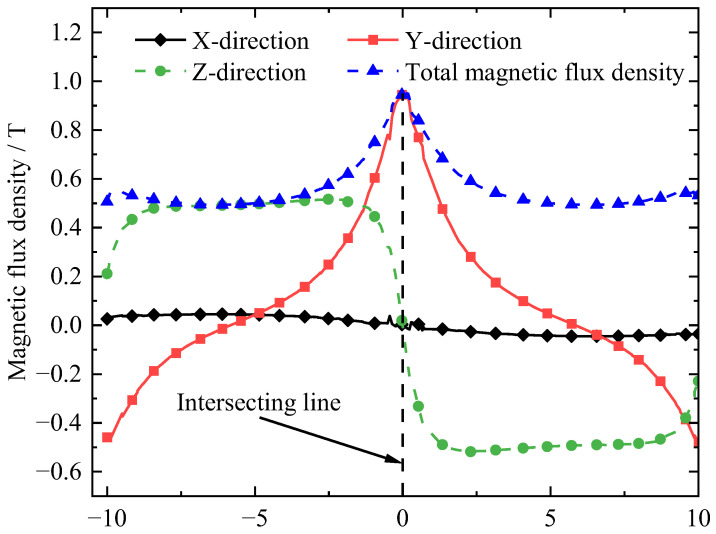
Magnetic flux density distribution in the X, Y and Z directions.

**Figure 3 micromachines-14-01242-f003:**
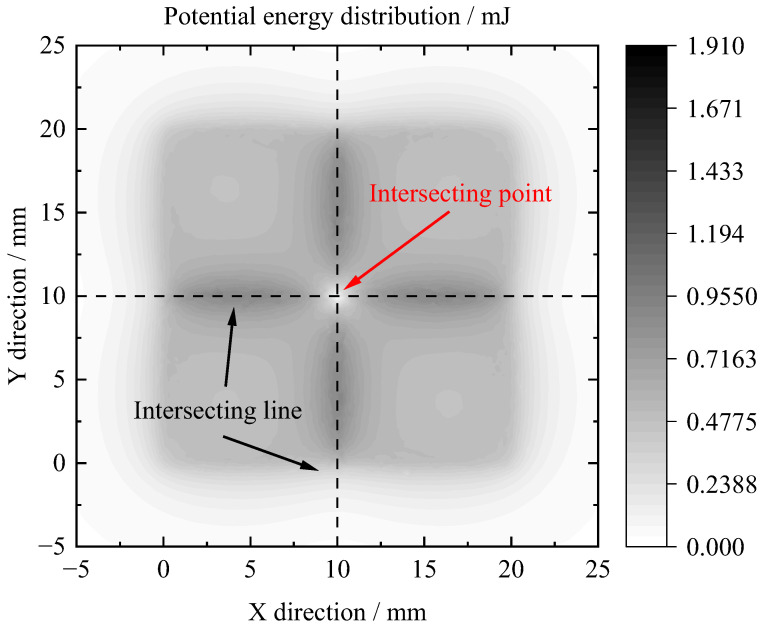
Distribution of potential energy above the PM array.

**Figure 4 micromachines-14-01242-f004:**
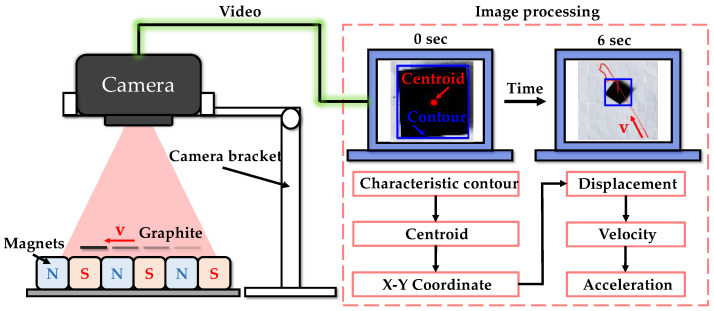
Conceptual diagram of the experimental system for in-plane motion.

**Figure 5 micromachines-14-01242-f005:**
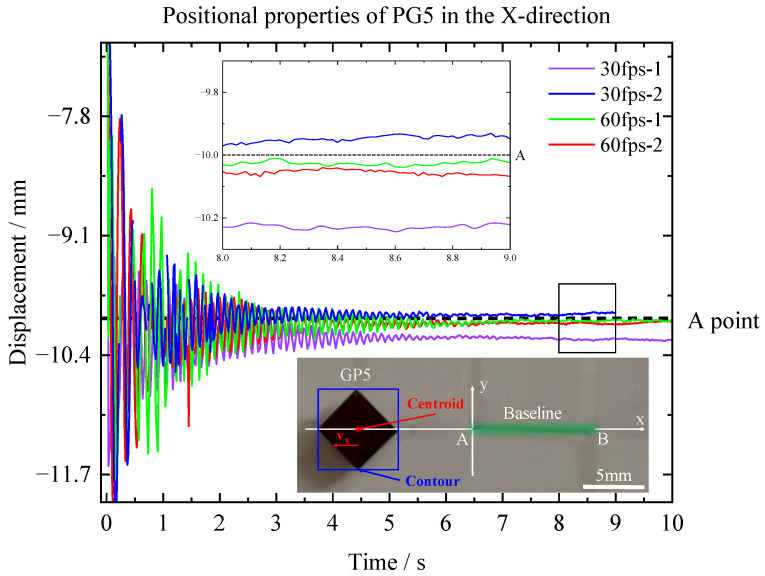
Potential properties of PG5 n the X-direction.

**Figure 6 micromachines-14-01242-f006:**
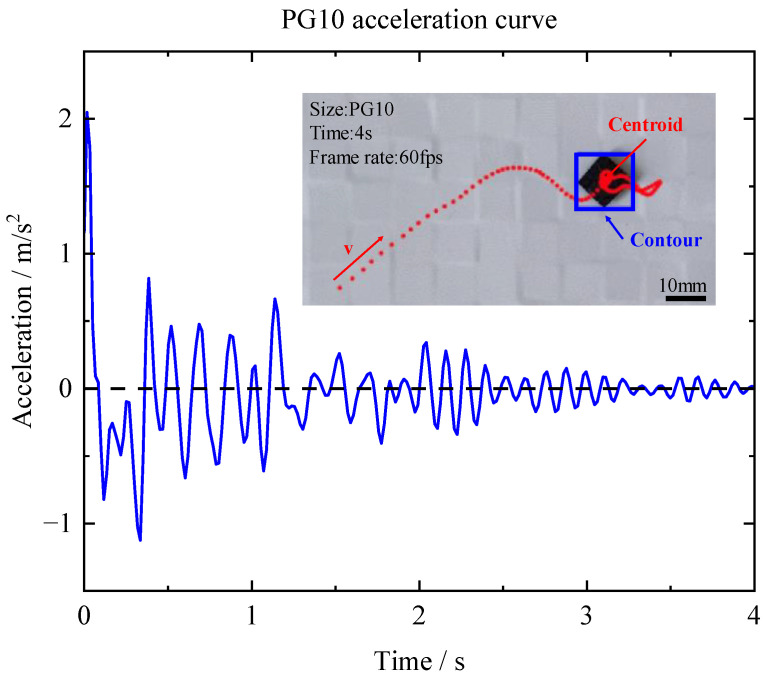
Acceleration of the in-plane motion of PG10.

**Figure 7 micromachines-14-01242-f007:**
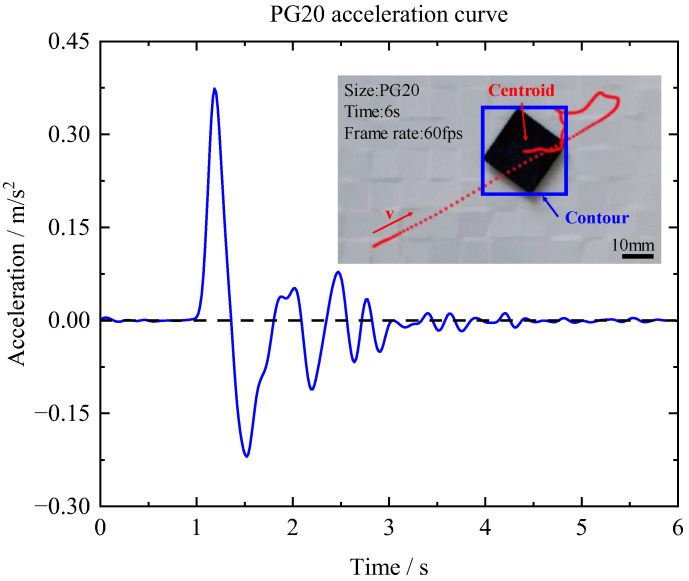
Acceleration of the in-plane motion of PG20.

**Figure 8 micromachines-14-01242-f008:**
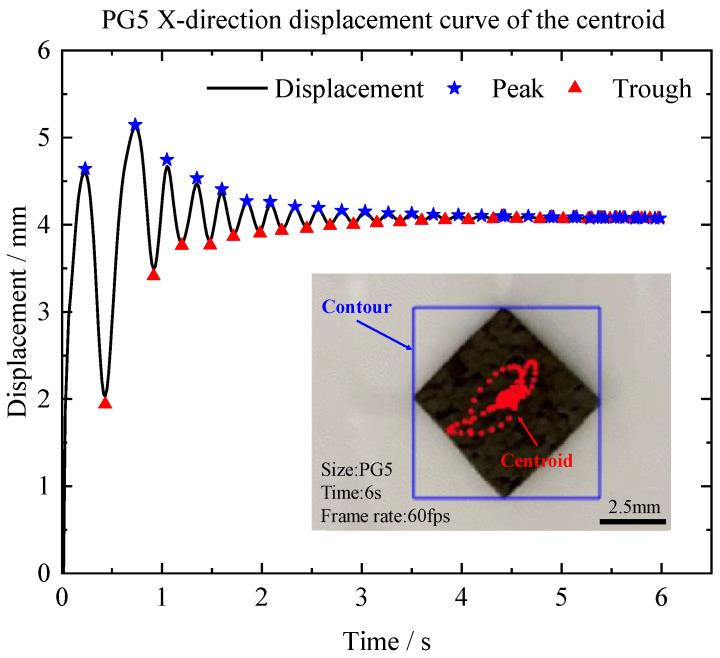
Oscillation displacement of motion in the X-direction for PG5.

**Figure 9 micromachines-14-01242-f009:**
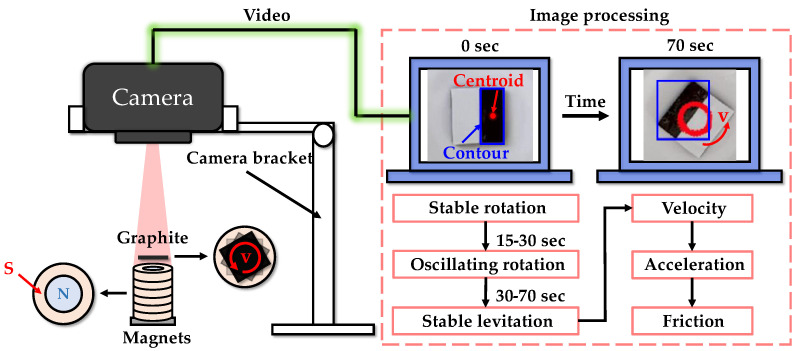
Conceptual diagram of the experimental system for fixed-axis rotation.

**Figure 10 micromachines-14-01242-f010:**
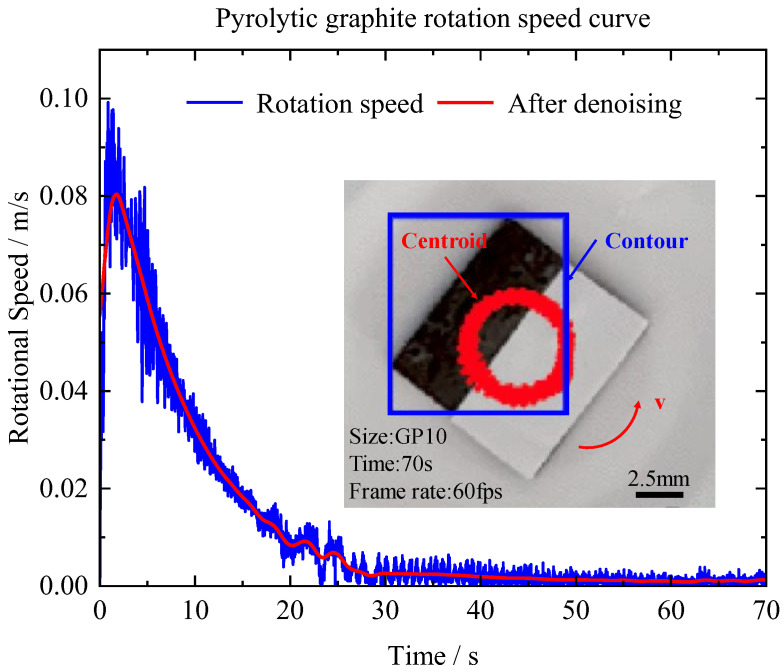
Rotational speed of fixed-axis rotation.

**Figure 11 micromachines-14-01242-f011:**
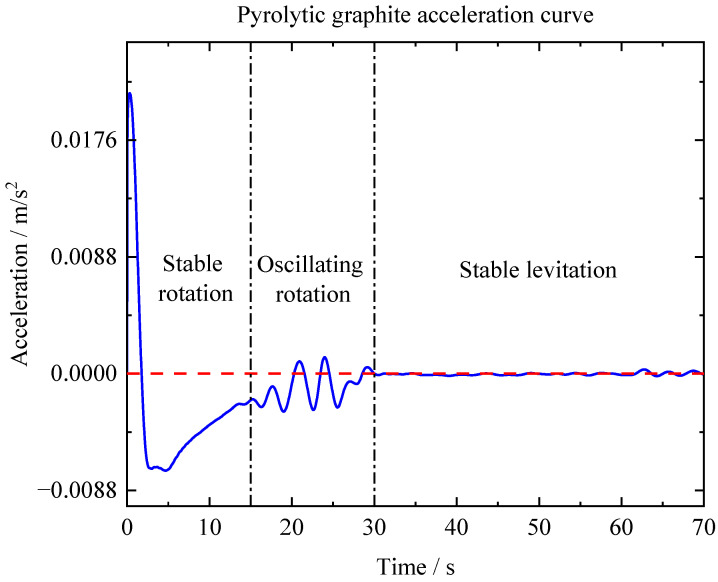
Acceleration of the stable rotation, oscillating rotation, and stable levitation stages.

**Table 1 micromachines-14-01242-t001:** In-plane forces and damping ratios for the in-plane motion of pyrolytic graphite.

Size	In-Plane Force/µN	Damping Ratio
PG5	-	ξ_5x_ = 0.166
		ξ_5y_ = 0.122
PG10	61.4	ξ_10x_ = 0.173
		ξ_10y_ = 0.142
PG20	28.2	-

**Table 2 micromachines-14-01242-t002:** Applications of the diamagnetic levitation of pyrolytic graphite with different size specifications.

Form of in-Plane Motion	Size	Potential Applications
In-plane translation	Smaller size	Precise positioning, drug delivery
	Larger size	Magnetic detection, crack detection
Fixed-axis rotation	-	Gas bearing, gas flow detectors

## Data Availability

Not applicable.

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
