# Peer review of "Pyrolytic Graphite for an In-Plane Force Study of Diamagnetic Levitation: A Potential Microdetector of Cracks in Magnetic Material"

_micromachines, 2023, doi:10.3390/mi14061242_

Round 1

Reviewer 1 Report

Comments:

The authors have written an article entitled as ‘Pyrolytic graphite for in-plane force study of diamagnetic levitation: a potential magnetic material cracks micro-detector.’. Unfortunately, this article is NOT well-organized scientific article, and consequently, it suffers from significant drawbacks, and some of them are listed as listed below:

1.      Usually, an abstract of any reader-friendly, scientific article is the ‘self-dependent and concise summary’ of the whole investigation; this is a MANDATORY criterion of writing an abstract regardless the types of article. However, the current version of the abstract does not satisfy the criterion, as stated above. This is a shortcoming of this article.

2.      The next important and most vital part of an article is the ‘introduction’ part. It is generally treated as the ‘heart’ of any scientific article. The introduction part usually guides the flow of the rest of the article’s parts. So, a question naturally arises on how to construct the introduction? It is a significant event for a good and well-organized scientific article. To address the above question, it is required to survey the existing literature on the subject matter of the article extensively, and this kind of literature survey will help author(s) to reveal the ‘research gap or originality’ within the existing literature. Once the ‘research gap’ is identified, then the rest of the article MUST be devoted to filling up the ‘research gap’ systematically order as identified. Regrettably, the ‘introduction’ of this article has not written as highlighted above. Without revealing the ‘research gap’ systematically, any sorts of research have no scientific value in reality! Hence, this is another severe shortcoming of this article!

3.      The quality of figures 1, 3-10 must be improved and current version figure 2 must be replaced by the colour version.  This is another shortcoming for this article.   

4.      The conclusion part MUST also be precise and straightforward as an abstract so that the potential readers can easily understand the subject matter. Again, the ‘conclusion’ part is not written as expected. Besides, the ‘conclusion’ must have consistency with the abstract; this is a common practice of writing a reader-friendly scientific article.

5.       The discussion should be omitted, and ‘Results and discussion’ section must be added.  Just after the conclusion part, an overall all future direction or suggestion for this research is mandatory for young researchers and this is one of the obligations for the authors of this article.   

6.      This article has much more peculiarities, and it is the authors’ responsibility to figure out all of them and address them accordingly.

 Anyway, please wait for the comments from the editorial office.

 The moderated English is necessary.

Author Response

Dear Editors and Reviewers:

Thank you for your letter and for the reviewers’ comments concerning our manuscript entitled “Pyrolytic graphite for in-plane force study of diamagnetic levitation: a potential magnetic material cracks micro-detector” (micromachines-2414980). Those comments are very valuable and very helpful for revising and improving our paper, as well as the important guiding significance to our researches. We have studied comments carefully and have made correction which we hope meet with approval. Revised portion are marked in red in the paper. Please refer to the “cover letter & respons letter” for our responses to each of the reviewer's comments.

Yours sincerely,

Mr. Runze Liu

26, May, 2025

School Of Astronautics, Beihang University 

Reviewer 2 Report

In the attached file, authors can find a review of their article.

Author Response

(The authors gave the same response as above.)

Reviewer 3 Report

The authors study the diamagnetic levitation process of pyrolytic graphite on an array of permanent magnets.   They demonstrate potential applications in precise positioning and non-destructive material testing.  The paper is well written, and easy to understand.  The authors substantiate their findings by an extensive theoretical treatment.  I think the paper does deserve publication.  I could not find anything requiring modification or revision, thus I think the paper can be published as is.

Author Response

(The authors gave the same response as above.)

Round 2

Reviewer 1 Report

Please wait for the comments from the editorial office. 

The quality of English is reasonable! 

Author Response

Cover letter

 Dear editor,

On behalf of my co-authors, we thank you very much for giving us an opportunity to revise our manuscript. We appreciate editor and reviewers very much for their positive and constructive comments and suggestions on our manuscript entitled “Pyrolytic graphite for in-plane force study of diamagnetic levitation: a potential magnetic material cracks micro-detector” (micromachines-2414980). We are very sorry to update the revised manuscript so late because we have improved and modified the problematic images.

In this revised version, we have addressed the concerns of the reviewers. An item-by-item response to there viewers' comments is enclosed, and the revision was marked in red fronts in the manuscript.

We appreciate for Editors/Reviewers’ warm work earnestly and hope that the correction will meet with approval. If there are any other modifications we could make, we would like very much to modify them and we really appreciate your help. Looking forward to hearing from you soon.

Yours sincerely,

Mr. Runze Liu

June 5, 2023

School Of Astronautics, Beihang University
